# Socio-demographic, maternal, and infant characteristics associated with pacifier use among six-months old infants in Clark County, Nevada

**Kaelia Lynn Saniatan**[1], **Smriti Neupane**[1], **Chad Cross**[2], **Gabriela Buccini**[1]*

**1** Department of Social and Behavioral Health, School of Public Health, University of Nevada, Las Vegas, Nevada, United States of America, **2** Department of Epidemiology & Biostatistics, School of Public Health, University of Nevada, Las Vegas, Nevada, United States of America

* gabriela.buccini@unlv.edu

**Data Availability Statement:** A de-identified data set is not possible to provide due to ethical and legal considerations. These sharing restrictions are imposed by the UNLV Institutional Board Review

## Abstract

### Background

Pacifier use can interfere with nurturing care practices such as breastfeeding, soothing, and sleeping. Due to contradicting beliefs, recommendations, and the high frequency of pacifier use, understanding its associations may support shaping equitable public health recommendations. This study explored the socio-demographic, maternal, and infant characteristics associated with pacifier use among six-months old infants in Clark County, Nevada.

### Method

Cross-sectional survey was conducted in 2021 with a sample of mothers (n = 276) of infants under six months old in Clark County, Nevada. Participants were recruited through advertisements in birth, lactation, pediatric care centers, and social media. We used binomial and multinomial logistic models to assess the association between pacifier use and the age of pacifier introduction, respectively, with household, maternal, infant, healthcare characteristics, and feeding and sleeping practices.

### Results

More than half of the participants offered pacifiers (60.5%). Pacifier use was higher among low-income households (OR (95% CI) 2.06 (0.99–4.27)), mothers who identified as non-Hispanic (OR (95% CI) 2.09 (1.22–3.59)), non-first-time mothers (OR (95% CI) 2.09 (1.11–3.05)), and bottle-feeding infants (OR (95% CI) 2.76 (1.35–5.65)). Compared to those who did not introduce a pacifier, non-Hispanic mothers (RRR (95% CI) 2.34 (1.30–4.21)) and bottle-fed infants (RRR (95% CI) 2.71 (1.29–5.69)) had a higher risk of introducing pacifier within two weeks. Likewise, infants living in food insecure households (RRR (95% CI) 2.53 (0.97–6.58)) and mothers who have more than one child (RRR (95% CI) 2.44 (1.11–5.34)) had a higher risk of introducing a pacifier after two weeks.

(IRB). The authors declare that a de-identified data set from this study are available upon request directly to Dr. Buccini, assistant professor at UNLV (gabriela.buccini@unlv.edu) and/or to the UNLV IRB (irb@unlv.edu).

**Funding:** The authors received no specific funding for this work.

**Competing interests:** The authors have declared that no competing interests exist.

## Conclusion

Pacifier use is independently associated with maternal income, ethnicity, parity, and bottle feeding among six-month-old infants living in Clark County, Nevada. Household food insecurity increased the relative risk of introducing a pacifier after two weeks. Qualitative research on pacifier use among families with diverse ethnic/racial backgrounds is needed to improve equitable interventions.

## Background

Nurturing care is characterized as an environment that is responsive, emotionally supportive, sensitive to children's health and nutritional needs, and developmentally appropriate and stimulating, with opportunities for play and exploration, protecting children from adversities [1, 2]. Nurturing mutually responsive care practices such as breastfeeding, soothing, and safe sleeping are critical public health approaches that will influence young children's ability to self-regulate their emotions and actions, shape early childhood development outcomes, and reduce early disparities [3, 4].

Pacifiers have been used globally by at least two-thirds of parents during their infants' first year as an aide to modulate nurturing care practices assisting with soothing, sleeping, and feeding [5]. Despite the immediate perceived benefits of pacifier use, there is strong evidence of the negative consequences of its use in the short- and long-term for the child. Pacifier use is related to atypical child development of oral functions [6], increased incidence of acute otitis media and other infections [7–9], speech problems [10], malocclusion [11, 12], lower levels of intelligence [13–15], and more recently, with early life weight outcomes [16] and the development of unhealthy lifestyles in adulthood including smoking, overeating, and other compulsive disorders [17, 18]. Further, the use of pacifiers has been associated with shorter exclusive breastfeeding duration [19, 20]. However, the causal relationship has not been proven due to methodological shortcomings in trials investigating this relationship [21]. On the other hand, there is evidence that reducing pacifier use may improve exclusive breastfeeding rates [22].

Recommendations for pacifier use vary across different cultures and purposes. The American Academy of Pediatrics recommends breastfeeding [23] and using pacifiers to prevent Sudden Infant Death Syndrome (SIDS). Pacifiers may be introduced after breastfeeding is firmly established [24] and should be used for the first six months of an infant's life; after this age, the pacifier should be removed [9, 25]. By contrast, the American Academy of Pediatrics recommends implementing breastfeeding-supportive hospital practices, including avoiding pacifiers [23]. The World Health Organization recommends counseling mothers on the use and risks of pacifiers as part of the "Ten Steps to Successful Breastfeeding," upon which the Baby-Friendly Hospital Initiative (BFHI) has been based [26]. Thus, there is no straightforward recommendation for pacifier use and age of introduction as it needs to consider the potential benefits (e.g., SIDS) and risks (e.g., interference with exclusive breastfeeding).

Due to these contradicting belief systems, recommendations, and the high frequency of use, understanding the associations of pacifier use may support shaping public health recommendations and interventions that are mutually nurturing, responsive, and culturally appropriate. Evidence from low and middle-income countries shows that infant characteristics linked with pacifier use include low birth weight, under six months of age, not breastfed within the first hour, nor in the maternity ward, nor on-demand, and tea consumed on the first day at home [20, 27]. Caregiver's characteristics included younger caregivers, primiparous, lower education

level, low socioeconomic status, smoking history, working outside the home, maternal stress, and depression [18, 25]. However, all studies acknowledged were performed in countries outside of the U.S. To date, as far as we know, no study has investigated the determinants of pacifier use using a representative sample of infants. Therefore, this study aims to identify the socio-demographic, maternal, and infant characteristics associated with of pacifier use and age of pacifier introduction among six-months older infants living in Clark County, Nevada.

## Methods

### Study design

This secondary data analysis is based on cross-sectional survey data from the 2021 Early Responsive Nurturing Care (EARN) survey, which investigated a wide range of factors impacting nurturing care practices (i.e., breastfeeding, soothing, and sleeping) among infants under six months old living in Clark County (Boulder City, Henderson, Las Vegas, North Las Vegas, and Mesquite), Nevada.

The study protocol was approved by the Institutional Review Board of the University of Nevada Las Vegas (protocol 1767759–2). Participation in the study was voluntary, and written informed consent was obtained from all mothers for themselves and on behalf of their participating children. No incentives were provided for participation in this survey.

### Setting

As of 2022, Clark County, Nevada, has a total population of 2,350,206, with more than half of this population (54.0%) being White non-Hispanic, followed by those who identify as Hispanic/Latino (32.8%) and non-Hispanic Black (12.8%) [26]. In 2020, 18.5% of non-Hispanic Black, 14.5% of Hispanic/Latino, and 5.5% of non-Hispanic (i.e., Asian, Native Hawaiian/ Pacific Islander, Native American, Bi-Racial) families lived below the poverty level [28]. 35.7% of households in Clark County have an average household income under $49,999 and an average household size of 2.74 persons [26].

In the Centers for Disease Control and Prevention (CDC) 2022 Breastfeeding Report Card, the prevalence of exclusive breastfeeding in Nevada through six months was 22.3%, slightly lower than the national prevalence of 24.9% [29]. However, a median value of 75.4% of mothers in Clark County received early prenatal care, slightly lower than the national average of 77% [30]. Socio-cultural barriers present disadvantages within breastfeeding families through the lack of lactation services. Clark County only has two hospitals accredited in the Baby Friendly Hospital Initiative [31].

### Sampling and data collection

The sample size was designed to represent live births in Clark County. Therefore, the Southern Nevada Health District's vital records birth certificate file was used as a source for the sampling frame, where infants who were born alive to mothers' residents in Clark County in the past year (2020) were considered the sampling unit. In 2020, live births were estimated at 25,586 per year. Assuming a confidence level of 90% and an error of 5% and considering 50% completion, the minimum sample size calculated was 268 mother-infant dyads. Eligible participants were mothers 18 years or older with an infant under the age of six months living within the Clark County district, including the city of Las Vegas, North Las Vegas, Henderson, Mesquite, and Boulder City. If mothers were found not to meet the eligibility criteria, they were not allowed to move forward with the survey.

A convenience sampling technique was employed. Participants were recruited through advertisements across birth, lactation, and pediatric care centers throughout Clark County, social media (i.e., Facebook) posts in groups of mothers living in the selected area, and paid advertisements. The 2021 EARN surveys were available in English and Spanish and were distributed only online due to COVID-19 safety measures. Consent was received before the start of the survey, and participants could stop the study at any time. Data collection started in August 2021 and ended in October 2021. There were 323 respondents, out of which 47 [14.6%] did not answer the survey questions regarding pacifier use and were excluded. This yielded an analytical sample of 276 participants to explore the associations of pacifier use.

## Measurements

**Outcomes.** Two dependent variables were defined: (1) Pacifier use and (2) Age of Pacifier Introduction.

1. Pacifier use. The key dependent variable was pacifier use in the previous 24 hours to complete the survey. The status of pacifier use is aimed at minimizing possible biases resulting from the informant's memory, which is recommended by the World Health Organization (WHO) when collecting information on nurturing care practices [32]. Pacifier use status was determined by the question, "In the last 24 hours, has your baby used a pacifier?" The response options were "yes" or "no."

2. Age of Pacifier Introduction. The second dependent variable was the age of pacifier introduction. The age of introduction was determined by the question, "When did you start giving pacifiers for your baby?" The response options were "No Pacifier Use," "Within Two Weeks of Life," or "After Two Weeks of Life."

**Co-variables.** Covariate selection was guided by the hierarchical framework [33] developed based on a literature review supporting associations with pacifier use [17, 18] and data available on the 2021 EARN survey. These determinations informed the hierarchical framework illustrating potential associations of pacifier use organized across five categories of variables: household characteristics (model 1), maternal characteristics (model 2), infant characteristics (model 3), healthcare characteristics (model 4), and infant feeding and sleeping practices (model 5) (Fig 1).

Household characteristics included household food insecurity screening (yes/no) and household income (≤$50,000/$50–79,999/$80–99,999/≥$100,000). For the household food insecurity screening, the Hunger Vital Sign™, a validated two-question screening tool based on the U.S. Household Food Security Survey Module, was used [33, 34]. Households were identified as being at risk for food insecurity if they answered either the two-question screening as 'often true' or 'sometimes true' (vs. 'never true'). The maternal characteristics included maternal age (18-24/25-44), maternal education (no college degree/college degree), maternal ethnicity (Hispanic/non-Hispanic; due to limited sample for logistic regression, the non-Hispanic category included non-Hispanic White/non-Hispanic Black /non-Hispanic Other (i.e., Asian, Native Hawaiian/Pacific Islander, Native American, Bi-Racial)), first-time mother (yes, primipara/no, multipara), and depression screening (low/high). For the depression screening, the modified two-item patient health questionnaire (PHQ-2) used [35]. Participants responding "nearly every day" or "3" to the questions were classified as high risk for depression. The infant characteristics included sex (male/female), low birth weight (yes/no), type of delivery (vaginal/c-section), and age of pacifier introduction (no pacifier use/within two weeks of life/after two weeks of life). The healthcare characteristics included whether the infant was delivered in a

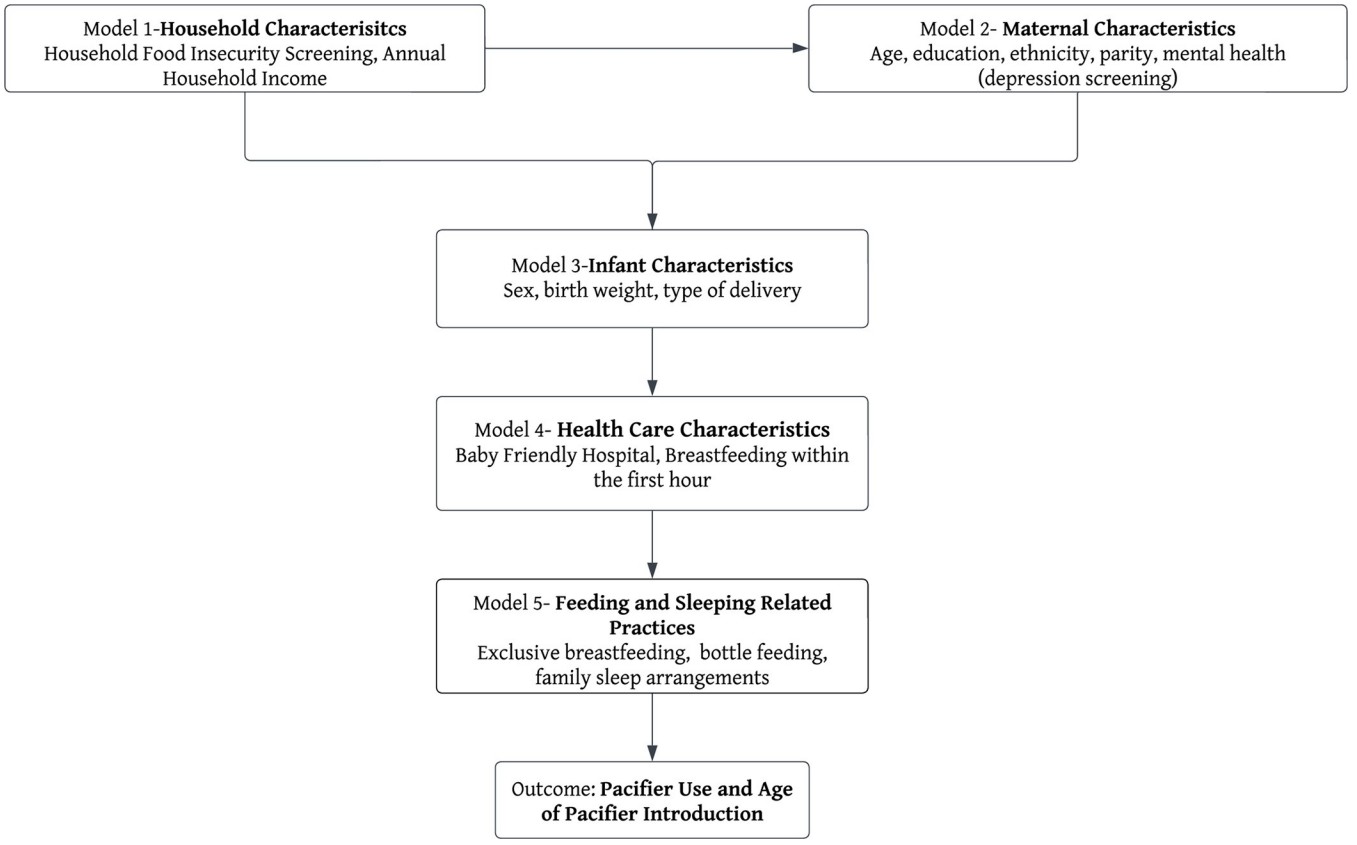

**Fig 1. Framework displaying the models used and their proximal relationship to the outcome.**

Baby-Friendly hospital accredited through the Baby Friendly Hospital Initiative (yes/no) and if the baby was breastfed within the first hour of life (yes/no). The infant feeding and sleeping practices included bottle-feeding (yes/no), exclusive breastfeeding (yes/no), and night-time family sleeping arrangements (bed sharing/no bed-sharing). As recommended by the WHO, infant feeding practices were determined by the status, i.e., on the day before the survey, to minimize possible biases resulting from the informant's memory. Exclusive breastfeeding was defined as recommended by the WHO [32]—children younger than six months who had received breast milk as their only source of nutrition and hydration, without any solid or liquid supplement, including water and tea. Exclusive breastfeeding information was confirmed with questions about intake in the previous 24 hours of tea, juice, water, or other milk/infant for-mula and about intake of other foods like fruit or savory food. Bottle-feeding was defined as the children who are fed with any liquid (including breast milk) or semi-solid food from a bot-tle with nipple/teat, as recommended by the WHO [36].

## Data analysis

Analyses were conducted using the Statistical Package for Social Sciences Version 28 (SPSS) and Stata Version 17. Descriptive analyses of the outcomes and covariates were conducted. Then, bivariate analyses were conducted to examine the associations between pacifier use out-comes and covariates. Covariates were selected for inclusion in a multivariable model when the association had a p-value <0.20 in the bivariate analyses.

To identify the associations of pacifier use, a hierarchical binomial logistic regression with robust variance [37] was performed to generate an adjusted odds ratio (AOR) and corresponding 95% confidence intervals (CIs). Similarly, to identify the associations of the age of pacifier introduction, a hierarchical multinomial logistic regression with robust variance to generate relative risk ratios (RRR) comparing the introduction of pacifier "Within Two Weeks," "After Two Weeks" to the reference "No Pacifier Use." The theoretical model presented in Fig 1 guided the hierarchical modelling data analysis approach [33, 38], which are particularly appropriate for use in studying determinants of childhood health outcomes such as breastfeeding [39] and pacifier use [27] as in the case of our analysis.

The distal level consisted of the household characteristics variables and was the first to be included in the analysis (level 1 model) and remained as the control for the more proximal hierarchical models. The first intermediate level of maternal characteristics variables was included the model (level 2 model), which was adjusted by the distal level model and remained as the control for the subsequent models. The second intermediate level of infant characteristics variables was included in the subsequent model (level 3 model), which was adjusted for variables in the previous two more distal models and remained as the control for the subsequent models. The third intermediate level consisted of healthcare characteristics variables and was included in the subsequent model (level l 4 model), which was adjusted for variables in the more distal three levels and remained as the control for the subsequent models. Lastly, the proximal level consisted of infant feeding and sleeping practices variables was included in the final model (level 5 model), which was adjusted for variables in the more distal levels. At each level model, a p value <0.05 was used as a statistical significance criterion to assess the correlation between variables and outcomes [33]. All variables that had a p-value <0.20 in the bivariate analyses were included in both binomial and multinomial hierarchical models and maintained in the models regardless of losing significance, as these data provide important adjustments to the parameter estimates in the final model [33].

## Results

The analytical sample consisted of 276 children under six months old from mothers in Clark County. Of the total participants, 60.5% of the respondents reported the use of a pacifier and were offered a pacifier within the first two weeks of life (n = 128, 46.4%). Around 14.9% of respondents were at risk for food insecurity. The majority of mothers were aged 25–44 (n = 248, 89.9%), had a college degree (n = 202, 73.2%), were non-Hispanic (n = 194, 70.3%), were not first-time mothers (n = 151, 54.7%), and had a low score for depression screening (n = 260, 94.2%). Among infants, most were female (n = 150, 54.3%) and were born from vaginal deliveries (n = 194, 70.3%). Although fewer babies were born in a Baby Friendly Hospital (n = 55, 19.9%), many were still placed on the breast within the first hour of delivery (n = 215, 77.9%). More than half of the mothers were exclusively breastfeeding (n = 148, 53.6%), most mothers bottle-fed their infants (n = 187, 67.8%) at the time of the survey, and 19.3% reported bed-sharing as their sleeping arrangement (Table 1).

Pacifier use was more frequent among infants from households with a median income of less than $50,000 (n = 36, 70.6%) compared to those with incomes greater than $100,000 (n = 60, 55.1%). Mothers identified as non-Hispanic (n = 127, 65.5%), first-time mothers (n = 86, 65.5%), as well as infants that were delivered via cesarean section (n = 57, 69.5%), and were not breastfed within the first hour of delivery (n = 43, 66.4%) had higher frequencies of pacifier use compared to the comparison groups. Lastly, infants that are exclusively breastfed (n = 82, 55.4%), not bottle fed (n = 39, 43.8%), and share a bed with their caregiver (n = 27, 50.9%) have lower frequencies of pacifier use (Table 2). Regarding the age of pacifier

**Table 1. Descriptive analysis of pacifier use, household, maternal, infant, and health care characteristics, and infant feeding and sleeping practices, 2021.**

| Study Variables | Frequency (n = 276) | Percent (%) |
|---|---|---|
| **Pacifier Use** | | |
| No | 109 | 39.5 |
| Yes | 167 | 60.5 |
| **Pacifier Introduction Age** | | |
| No Pacifier Use | 109 | 39.5 |
| Within Two Weeks | 128 | 46.4 |
| After Two Weeks | 39 | 14.1 |
| **Household Income** | | |
| ≥$100,000 | 109 | 39.5 |
| $50,000–99,999 | 116 | 42.0 |
| ≤$50,000 | 51 | 18.5 |
| **Household Food Insecurity (n = 266)** | | |
| Food Secured | 226 | 85.0 |
| Food Insecure | 40 | 15.0 |
| **Maternal Age** | | |
| 18–24 | 28 | 10.1 |
| 25–44 | 248 | 89.9 |
| **Mother's Education** | | |
| College Degree | 202 | 73.2 |
| No College Degree | 74 | 26.8 |
| **Maternal Ethnicity** | | |
| Hispanic | 82 | 29.7 |
| Non-Hispanic | 194 | 70.3 |
| **First Time Mother** | | |
| Multiparous | 151 | 54.7 |
| Primiparous | 125 | 45.3 |
| **Depression Screening** | | |
| Low Risk | 260 | 94.2 |
| High Risk | 16 | 5.80 |
| **Child's Sex** | | |
| Female | 150 | 54.3 |
| Male | 126 | 45.7 |
| **Type of Delivery** | | |
| Vaginal | 194 | 70.3 |
| C-Section | 82 | 29.7 |
| **Baby Friendly Hospital** | | |
| Yes | 55 | 19.9 |
| No | 221 | 80.1 |
| **Breastfed in First Hour** | | |
| Yes | 215 | 77.9 |
| No | 61 | 22.1 |
| **Exclusive Breastfeeding** | | |
| Yes | 148 | 53.6 |
| No | 128 | 46.4 |
| **Bottle Feeding** | | |
| No | 89 | 32.2 |

(*Continued*)

**Table 1.** (Continued)

| Study Variables | Frequency (n = 276) | Percent (%) |
|---|---|---|
| Yes | 187 | 67.8 |
| **Sleeping Arrangements (n = 274)** | | |
| No Bed Sharing | 221 | 80.7 |
| Bed Sharing | 53 | 19.3 |

introduction, non-Hispanic mothers (n = 100, 51.5%) introduced pacifiers within two weeks, while more Hispanic mothers (n = 12, 14.6%) introduced them after two weeks. Families living in food-insecure households (n = 9, 22.5%) were more likely to introduce pacifiers after two weeks compared to food-secure households. Infants that were breastfed within the first hour of delivery had a lower prevalence of introduction both within two weeks (n = 94, 43.7%) and after two weeks (n = 30, 13.9%) compared to those who were not breastfed within the first hour. Lastly, infants who were not bottle-fed infants were less likely to be introduced to a pacifier within two weeks (n = 30, 33.7%) and after two weeks (n = 9, 10.1%) (Table 3).

Pacifier use was independently associated with mothers who identified as non-Hispanic (OR (95% CI) 2.09 (1.22–3.59)) and mothers who have more than one child (OR (95% CI) 2.09 (1.11–3.05])) in level 2, and with bottle-feeding infants (OR (95% CI) 2.76 (1.35–5.65)) in level 5 (Table 4).

Compared with those who did not introduce a pacifier, infants living in food insecure households (RRR (95% CI) 2.53 (1.00–6.58)) in level 1 and mothers who have more than one child (RRR (95% CI) 2.44 (1.11–5.34)) in level 2 had a higher risk of introducing a pacifier after two weeks. Likewise, non-Hispanic mothers (RRR (95% CI) 2.34 (1.30–4.21)) in level 2 and bottle-fed infants (RRR (95% CI) 2.71 (1.29–5.69)) in level 5 had a higher risk of introducing pacifier within two weeks (Table 5).

## Discussion

By taking a hierarchal modelling data analysis approach our study also identified that pacifier use is associated with maternal ethnicity and parity (level 2 model) and bottle feeding (level 5 model) among six-month-old living in Clark County, Nevada. In addition to these known factors, our study found that the age of pacifier introduction was associated with household food insecurity (level 1 model). There is limited research on the relationship between pacifier use, age of introduction, and food insecurity. Food insecurity, defined as "a household-level economic and social condition of limited or uncertain access to adequate food" [40], can cause nutrient deficiencies that can affect the health of both the caregiver and the child [41]. In turn, food insecurity is a chronic stressor that can lead to mental health issues for the caregiver, such as anxiety and depression, which can affect parenting skills and the ability to provide nurturing care to their infant [42]. An emotionally distressed caregiver may be more likely to introduce a pacifier [42, 43] rather than finding alternative soothing methods to calm a fussy baby. Although maternal depression did not sustain an association in the multivariate analysis, it is essential to consider when developing pacifier-use interventions [17, 20, 43, 44], especially among food-insecure families, which due to COVID is a group that has increased exponentially in recent years [45].

Our study highlights the role of maternal ethnicity in pacifier use practices. While mothers of Hispanic ethnicity were more likely not to offer a pacifier, mothers of non-Hispanic ethnicity were found to be more likely to offer a pacifier. Prior studies have linked maternal ethnicity with disparities in child health and nutrition outcomes, such as low exclusive breastfeeding

**Table 2. Bivariate analysis of pacifier use by household, maternal, infant, and health care characteristics, and infant feeding and sleeping practices, 2021.**

| Variables | Pacifier Use | | No Pacifier Use | | P-Value |
|---|---|---|---|---|---|
| | n | % | n | % | |
| **Household Income** | | | | | |
| ≥$100,000 | 60 | 55.1 | 49 | 44.9 | **.169**** |
| $50,000–99,999 | 71 | 61.2 | 45 | 38.8 | |
| ≤$50,000 | 36 | 70.6 | 15 | 29.4 | |
| **Household Food Insecurity (n = 266)** | | | | | |
| Food Secured | 131 | 58.0 | 95 | 42.0 | .258 |
| Food Insecure | 27 | 67.5 | 13 | 32.5 | |
| **Maternal Age** | | | | | |
| 18–24 | 18 | 64.3 | 10 | 35.7 | .666 |
| 25–44 | 149 | 60.1 | 99 | 39.9 | |
| **Mother's Education** | | | | | |
| College Degree | 118 | 58.4 | 84 | 41.6 | .240 |
| No College Degree | 49 | 66.2 | 25 | 33.8 | |
| **Maternal Ethnicity** | | | | | |
| Hispanic | 40 | 48.8 | 42 | 51.2 | **.010**** |
| Non-Hispanic | 127 | 65.5 | 67 | 34.5 | |
| **First Time Mother** | | | | | |
| Multiparous | 81 | 53.6 | 70 | 46.4 | **.010**** |
| Primiparous | 86 | 68.8 | 39 | 31.2 | |
| **Depression Screening** | | | | | |
| Low Risk | 156 | 60.0 | 104 | 40.0 | .487 |
| High Risk | 11 | 68.8 | 5 | 31.2 | |
| **Child's Sex** | | | | | |
| Female | 88 | 58.7 | 62 | 41.3 | .495 |
| Male | 79 | 62.7 | 47 | 37.3 | |
| **Type of Delivery** | | | | | |
| Vaginal | 110 | 56.7 | 84 | 43.3 | **.047**** |
| C-Section | 57 | 69.5 | 25 | 30.5 | |
| **Baby Friendly Hospital** | | | | | |
| Yes | 34 | 61.8 | 21 | 38.2 | .824 |
| No | 133 | 60.2 | 88 | 39.8 | |
| **Breastfed in First Hour** | | | | | |
| Yes | 124 | 57.7 | 91 | 42.3 | **.071**** |
| No | 43 | 70.5 | 18 | 29.5 | |
| **Exclusive Breastfeeding** | | | | | |
| Yes | 82 | 55.4 | 66 | 44.6 | **.062**** |
| No | 85 | 66.4 | 43 | 33.6 | |
| **Bottle Feeding** | | | | | |
| No | 39 | 43.8 | 50 | 56.2 | **< .001**** |
| Yes | 128 | 68.4 | 59 | 31.6 | |
| **Sleeping Arrangements (n = 274)** | | | | | |
| No Bed Sharing | 139 | 62.9 | 82 | 37.1 | **.110**** |
| Bed Sharing | 27 | 50.9 | 26 | 49.1 | |

**p<0.20

**Table 3. Bivariate analysis of the age of pacifier use introduction by household, maternal, infant, and health care characteristics, and infant feeding and sleeping practices, 2021.**

| Variables | Pacifier Introduced Within 2 Weeks | | Pacifier Introduced After 2 Weeks | | P-Value |
|---|---|---|---|---|---|
| | n | % | n | % | |
| **Household Income** | | | | | |
| ≥$100,000 | 48 | 44.0 | 12 | 11.0 | .273 |
| $50,000–99,999 | 55 | 47.4 | 16 | 13.8 | |
| ≤$50,000 | 25 | 49.0 | 11 | 21.6 | |
| **Household Food Insecurity (n = 266)** | | | | | |
| Food Secured | 105 | 46.5 | 26 | 11.5 | .142** |
| Food Insecure | 18 | 45.0 | 9 | 22.5 | |
| **Maternal Age** | | | | | |
| 18–24 | 13 | 46.4 | 5 | 17.9 | .811 |
| 25–44 | 115 | 46.4 | 34 | 13.7 | |
| **Mother's Education** | | | | | |
| College Degree | 90 | 44.6 | 28 | 13.9 | .493 |
| No College Degree | 38 | 51.3 | 11 | 14.9 | |
| **Maternal Ethnicity** | | | | | |
| Hispanic | 28 | 34.1 | 12 | 14.6 | .020** |
| Non- Hispanic | 100 | 51.5 | 27 | 13.9 | |
| **First Time Mother** | | | | | |
| Multiparous | 65 | 43.0 | 16 | 10.6 | .021** |
| Primiparous | 63 | 50.4 | 23 | 18.4 | |
| **Depression Screening** | | | | | |
| Low Risk | 120 | 46.1 | 36 | 13.8 | .742 |
| High Risk | 8 | 50.0 | 3 | 18.7 | |
| **Child's Sex** | | | | | |
| Female | 35 | 43.3 | 23 | 15.3 | .529 |
| Male | 63 | 50.0 | 16 | 12.7 | |
| **Type of Delivery** | | | | | |
| Vaginal | 87 | 44.8 | 23 | 11.9 | .077** |
| C-Section | 41 | 50.0 | 16 | 19.5 | |
| **Baby Friendly Hospital** | | | | | |
| Yes | 22 | 40.0 | 12 | 21.8 | .173** |
| No | 106 | 48.0 | 27 | 12.2 | |
| **Breastfed in First Hour** | | | | | |
| Yes | 94 | 43.7 | 30 | 13.9 | .176** |
| No | 34 | 55.7 | 9 | 14.7 | |
| **Exclusive Breastfeeding** | | | | | |
| Yes | 65 | 43.9 | 17 | 11.5 | .129** |
| No | 63 | 49.2 | 22 | 17.2 | |
| **Bottle Feeding** | | | | | |
| No | 30 | 33.7 | 9 | 10.1 | < .000** |
| Yes | 98 | 52.4 | 30 | 16.0 | |
| **Sleeping Arrangements (n = 274)** | | | | | |
| No Bed Sharing | 104 | 47.1 | 35 | 15.8 | .154** |
| Bed Sharing | 23 | 43.4 | 4 | 07.5 | |

**p<0.20

**Table 4. Hierarchical logistic regression on the association between pacifier use and household, maternal, infant, and health care characteristics as well as infant feeding and sleeping practices, 2021.** Each model level represents the addition of blocks of model variables as specified. A value of "1" across rows indicates the comparison group in the model categorical predictors.

| | Pacifier use Adjusted OR (95% CI) | | | | |
|---|---|---|---|---|---|
| | Level 1 Model | Level 2 Model | Level 3 Model | Level 4 Model | Level 5 Model |
| **Level 1 Variables: Household Characteristics** | | | | | |
| **Household Income** | | | | | |
| ≥$100,000 | **1** | 1 | 1 | 1 | 1 |
| $50,000–99,999 | **1.29 (0.76–2.19)** | 1.34 (0.77–2.33) | 1.29 (0.74–2.25) | 1.28 (0.74–2.23) | 1.52 (0.86–2.69) |
| ≤$50,000 | **1.96 (0.96–3.99)** | 2.06 (0.99–4.27) | 1.99 (0.96–4.15) | 1.95 (0.94–4.07) | 2.73 (1.22–6.10) |
| **Level 2 Variables: Maternal Characteristics** | | | | | |
| **Maternal Ethnicity** | | | | | |
| Hispanic | - | **1** | 1 | 1 | 1 |
| Non-Hispanic | - | **2.09* (1.22–3.59)** | 2.04 (1.18–3.50) | 2.02 (1.18–3.49) | 2.16 (1.23–3.77) |
| **Parity** | | | | | |
| Primiparous | - | **1** | 1 | 1 | 1 |
| Multiparous | - | **1.84* (1.11–3.05)** | 1.78 (1.07–2.96) | 1.72 (1.02–2.90) | 1.37 (0.77–2.43) |
| **Level 3 Variables: Infant Characteristics** | | | | | |
| **Type of Delivery** | | | | | |
| Vaginal | - | - | **1** | 1 | 1 |
| C-section | - | - | **1.52 (0.86–2.70)** | 1.45 (0.82–2.58) | 1.40 (0.76–2.56) |
| **Level 4 Variables: Healthcare Characteristics** | | | | | |
| **Breastfed in First Hour** | | | | | |
| Yes | - | - | - | **1** | 1 |
| No | - | - | - | **1.23 (0.64–2.37)** | 1.24 (0.62–2.50) |
| **Level 5 Variables: Infant Feeding and Sleeping Practices** | | | | | |
| **Exclusive Breastfeeding** | | | | | |
| Yes | - | - | - | - | **1** |
| No | - | - | - | - | **0.83 (0.43–1.59)** |
| **Bottle Feeding** | | | | | |
| No | - | - | - | - | **1** |
| Yes | - | - | - | - | **2.76* (1.35–5.65)** |
| **Sleeping Arrangements** | | | | | |
| No Bed Sharing | - | - | - | - | **1** |
| Bed Sharing | - | - | - | - | **0.68 (0.35–1.34)** |

*p<0.05

rates [46–48], high infant mortality [49], and lower usage of emergency department services [50]. However, there is a lack of data exploring the different ways maternal ethnicity influences pacifier use across diverse settings, which may contribute to the lack of culturally appropriate nurturing care support amplifying disparities among communities of color [48, 51]. We acknowledge that our data is limited to compare maternal ethnicity and future studies should analyze the influence of maternal race on pacifier outcomes. Maternal parity throughout research has been shown to influence an increase in pacifier use and early cessation of breastfeeding [27, 52]. The research aligns with the data displayed in this study. A study in Australia investigated why a first-time caregiver may offer their infant a pacifier. It was found that an area of opportunity to educate first-time mothers on pacifier use is with their families. Many first-time mothers received pacifier use advice from their own mothers and/or mothers-in-law [52]. Therefore, when developing an intervention to reduce pacifier use, involving other

**Table 5. Multinomial logistic regression on the association between age of pacifier use introduction and household, maternal, infant, and health care characteristics as well as infant feeding and sleeping practices, 2021.** A value of "1" across rows indicates the comparison group in the model categorical predictors.

| | Age of pacifier use introduction RRR (95% CI) | | | | | | | | | |
|---|---|---|---|---|---|---|---|---|---|---|
| | Level 1 Model | | Level 2 Model | | Level 3 Model | | Level 4 Model | | Level 5 Model | |
| | Pacifier Introduced Within 2 Weeks | Pacifier Introduced After 2 Weeks | Pacifier Introduced Within 2 Weeks | Pacifier Introduced After 2 Weeks | Pacifier Introduced Within 2 Weeks | Pacifier Introduced After 2 Weeks | Pacifier Introduced Within 2 Weeks | Pacifier Introduced After 2 Weeks | Pacifier Introduced Within 2 Weeks | Pacifier Introduced After 2 Weeks |
| Level 1 Variables: Household Characteristics | | | | | | | | | | |
| **Household Food Insecurity** | | | | | | | | | | |
| Food Secured | **1** | **1** | 1 | 1 | 1 | 1 | 1 | 1 | 1 | 1 |
| Food Insecure | **1.25 (0.58–2.70)** | **2.53* (1.00–6.58)** | 1.37 (0.64–2.93) | 2.80 (1.06–7.42) | 1.39 (0.64–2.99) | 2.95 (1.10–7.87) | 1.33 [0.61–2.90] | 3.25 (1.19–8.88) | 1.46 (0.65–3.28) | 3.45 (1.22–9.71) |
| Level 2 Variables: Maternal Characteristics | | | | | | | | | | |
| **Maternal Ethnicity** | | | | | | | | | | |
| Hispanic | - | | **1** | **1** | 1 | 1 | 1 | 1 | 1 | 1 |
| Non-Hispanic | - | | **2.34* (1.30–4.21)** | **1.56 (0.66–3.67)** | 2.28 (1.27–4.12) | 1.44 (0.61–3.41) | 2.32 (1.27–4.23) | 1.32 (0.55–3.18) | 2.42 (1.31–4.48) | 1.27 (0.51–3.13) |
| **Parity** | | | | | | | | | | |
| Primiparous | - | | **1** | **1** | 1 | 1 | 1 | 1 | 1 | 1 |
| Multiparous | - | | **1.65 (0.96–2.84)** | **2.44* (1.11–5.34)** | 1.62 (0.94–2.78) | 2.33 (1.06–5.14) | 1.56 (0.90–2.71) | 2.36 (1.04–5.33) | 1.31 (0.72–2.39) | 1.96 (0.87–4.42) |
| Level 3 Variables: Infant Characteristics | | | | | | | | | | |
| **Type of Delivery** | | | | | | | | | | |
| Vaginal | - | | - | - | **1** | **1** | 1 | 1 | 1 | 1 |
| C-section | - | | - | - | **1.38 (0.76–2.53)** | **2.10 (0.91–4.87)** | 1.28 (0.69–2.38) | 2.33 (0.97–5.57) | 1.27 (0.66–2.43) | 2.29 (0.96–5.42) |
| Level 4 Variables: Healthcare Characteristics | | | | | | | | | | |
| **Baby Friendly Hospital** | | | | | | | | | | |
| Yes | - | | - | - | - | - | **1** | **1** | 1 | 1 |
| No | - | | - | - | - | - | **1.31 (0.62–2.73)** | **0.51 (0.21–1.25)** | 1.31 (0.63–2.74) | 0.47 (0.18–1.20) |
| **Breastfed in First Hour** | | | | | | | | | | |
| Yes | - | | - | - | - | - | **1** | **1** | 1 | 1 |
| No | - | | - | - | - | - | **1.34 (0.65–2.64)** | **0.74 (0.25–2.17)** | 1.38 (0.68–2.80) | 0.79 (0.26–2.36) |
| Level 5 Variables: Infant Feeding and Sleeping Practices | | | | | | | | | | |
| **Exclusive Breastfeeding** | | | | | | | | | | |
| Yes | - | | - | - | - | - | - | - | **1** | **1** |
| No | - | | - | - | - | - | - | - | **0.75 (0.38–1.46)** | **0.99 (0.38–2.60)** |
| **Bottle Feeding** | | | | | | | | | | |
| No | - | | - | - | - | - | - | - | **1** | **1** |
| Yes | - | | - | - | - | - | - | - | **2.71* (1.29–5.69)** | **1.98 (0.69–5.65)** |
| **Sleeping Arrangements** | | | | | | | | | | |

*(Continued)*

**Table 5.** (Continued)

| | Age of pacifier use introduction RRR (95% CI) | | | | | | | | | |
|---|---|---|---|---|---|---|---|---|---|---|
| | Level 1 Model | | Level 2 Model | | Level 3 Model | | Level 4 Model | | Level 5 Model | |
| | Pacifier Introduced Within 2 Weeks | Pacifier Introduced After 2 Weeks | Pacifier Introduced Within 2 Weeks | Pacifier Introduced After 2 Weeks | Pacifier Introduced Within 2 Weeks | Pacifier Introduced After 2 Weeks | Pacifier Introduced Within 2 Weeks | Pacifier Introduced After 2 Weeks | Pacifier Introduced Within 2 Weeks | Pacifier Introduced After 2 Weeks |
| No Bed Sharing | - | | - | - | - | - | - | - | 1 | 1 |
| Bed Sharing | - | | - | - | - | - | - | - | 0.90 (0.45–1.82) | 0.50 (0.15–1.73) |

*p<0.05; reference category is no pacifier use.

members of the family such as fathers and grandmothers has shown to be effective [53], further demonstrating why culturally appropriate approaches are imperative in creating equitable solutions.

Corroborating our findings, bottle feeding has been associated with increased pacifier use in infants [54]. Increased use of bottle feeding has been associated with low birth weight, whether this was the first child, delivery by cesarean section, and male sex [25]. Bottle feeding along with pacifier use can create difficulties with breastfeeding compared to infants not offered an artificial nipple [54]. Introducing an artificial nipple, by either bottle or pacifier use, can create "nipple confusion," which refers to an infant's inability to establish proper oral configuration, latching techniques, and suck patterns to extract milk from the breast after being exposed to an artificial nipple [55]. Nipple confusion can occur due to the inability of some infants to adapt to different oral configurations, the "imprinting" in latching learning that occurs in the neonatal period, and initial difficulties in latching on to the breast, who are more susceptible to nipple confusion. A literature review found robust evidence of nipple confusion due to bottle use since it releases milk faster than sucking the breast and the use of both pacifiers and bottles [55].

Pacifier use is still controversial due to its implied benefits and drawbacks. In contrast, the use of a pacifier is recommended for specific outcomes, such as the protection against Sudden Infant Death Syndrome (SIDS) [24], stimulation of non-nutritive sucking [43], pain management in the newborn [56], and regulation of fussy behavior [57], the extended use of a pacifier can lead to further health disparities. Disparities include but are not limited to the poor development of oral functions [6], increased incidence of acute otitis media, and other infections [7–9], malocclusion [11, 12], early life weight outcomes [16], the development of unhealthy lifestyles in adulthood including smoking, overeating, and other compulsive disorders [17, 18] as well as pacifier use beyond six months, have been associated with accelerated infant growth and toddler obesity [14]. Moreover, a pacifier use has been associated to speech problems [10], lower social interactions [58], and lower levels of intelligence. A recent analysis of a birth cohort study found a strong association between pacifier use and lower intelligence quotient at six years [14]. One possible hypothesis is that children using a pacifier, especially those who use it more intensely, are less stimulated affecting social interaction skills and may lose a critical window for optimal early childhood development exacerbating disparities early in life.

Addressing these disparities is one of the overarching goals of all areas of public health to create a healthier population [59]. In addition, pacifier use may contribute to disparities in exclusive breastfeeding. However, the causal relationship has not been fully elucidated [20], it is known that promoting exclusive breastfeeding can reduce pacifier use [22]. In the context of

our study, the prevalence of exclusive breastfeeding in Nevada through six months was 22.3% [29], which is slightly lower than the national prevalence, but far beyond the Global Maternal, Infant, and Young Child Nutrition Target to reach at least 70.0% of EBF by 2030 [59]. Therefore, the exclusive breastfeeding goal in Nevada will not be met unless key modifiable risk factors such as pacifier use are culturally appropriately addressed [20]. Some evidence-based initiatives that could embed culturally appropriate messages to promote breastfeeding and reduce pacifier use may include (i) providing education programs for parents and caregivers through various channels such as prenatal classes, postnatal support groups, and online resources [60], (ii) increasing access to community-based initiatives that offer perinatal support through peer counselor [61] and lactation consultants [62], (iii) training of health providers can help ensure that parents receive consistent messages on pros and cons of pacifier use, and (iv) promoting regulatory policies such as restrictions of predatory marketing of pacifiers [63, 64]. Many of these recommendations are embedded into the Baby Friendly Hospital Initiative which follows the "Ten Steps to Successful Breastfeeding". Steps 8 & 9 of the "Ten Steps. . ." focus on educating parents to recognize and respond to their infant's feeding cues and the pros and cons associated with pacifier use, respectively. While our study did not find an association between place of birth and pacifier use, potentially due to the low coverage of births occurred in Baby Friendly Hospitals in Clark County (19.9% vs 27.0% in the U.S–[31]); it is important to note that scaling up Baby Friendly Hospital Initiative has been associated with increased rates of exclusive breastfeeding and decreased rates of pacifier use [65].

Our study has some limitations to be considered when generalizing the findings. We surveyed a convenience sample of infants under six months old across Clark County, Nevada. Several data collection efforts across birth, lactation, and pediatric care centers throughout Clark County were made to recruit a diverse population of mothers; however, due to the COVID-19 pandemic, the majority of the 2021 EARN surveys' sample was recruited through paid advertisement on social media. We acknowledge that this may have limited the diversity of the sample size; however, our data were similar compared to the demographic data (i.e., ethnicity, educational attainment, and household income) in Clark County [66]. Nevertheless, our findings may be generalized to U.S. areas with a similarly high proportion of urban populations as Clark County.

To prevent recall bias questions pertaining to pacifier use were formatted to capture the status of the use on the 24-h prior to the survey as recommended by WHO [32]. In doing so, we understand that this may have limited our results by excluding mother-infant dyads that used a pacifier until certain age in infancy. On the other hand, it minimizes the participant to provide false or inaccurate answers. Another limitation is that we did not collect the age of the infant at the time of the survey. We acknowledge that practices are influenced by the infant's age, and some associations may not be found or weakened due to lack of age detail. The intensity of pacifier uses and psychosocial factors that may influence the breastfeeding process, including the infant's behavior (e.g., temperament and the mother's breastfeeding intentions), were not collected. Due to the cross-sectional design, the temporal sequence of events between pacifier use and the associated factors cannot be established; while reverse causality cannot be ruled out, the use of a conceptual hierarchical approach considering social, biological, and temporal relationships may leverage complex inter-relationships between these determinants [33]. Further, cross-sectional designs can generate hypotheses for the development of longitudinal studies [67].

The selection of a hierarchical approach to analyze the data come with both advantages and limitations throughout this study. The use of the hierarchical model was selected for its ability to measure significance across multiple variables in relation to their level influence to the outcomes [33]. Many of the selected variables in this study are interrelated to each other as they

can influence the overall significance to the outcomes. The hierarchical display of variables showcases these complex inter-relationships [33]. Due to this, some variables would display significance in one level of analysis while losing significance when other levels are added to the model. Regardless, variables were maintained in the entire model despite having lost statistical significance after the inclusion of inferior variables as they provide insight to the adjustments made in the analysis [27].

In this context, our findings support the importance to promote evidence-based initiatives that embed culturally appropriate messages to reduce pacifier use, including the scaling up of Baby Friendly Hospitals and community-based education programs for parents and caregivers. Hence, recommendations regarding pacifier use should use a counseling approach to support conscious decisions considering each family's context, culture, and goals regarding nurturing care practices, including exclusive breastfeeding goals [19–22]. Messages should be delivered by taking a welcoming and listening approach that encompasses both standardized information regarding the pros and cons of pacifier use, early childhood developmental expected behaviors, soothing methods to calm a fussy baby can help parents to understand and respond assertively to their child's needs [57]; by favoring dialogic communication, emotions that generate guilt, pain, social pressure, and obligation, which are not infrequent, can be welcomed and contribute to reflecting on culturally appropriate nurturing care resources to help reduce disparities in breastfeeding and pacifier use practices. Lastly, tailored messages for families who choose to offer the pacifier, including options to limited by one-year-old or restricted use at critical times and, once the habit of using a pacifier is established, support those families for withdrawal are critical [68]. Further qualitative research into cultural aspects of pacifier use among families from diverse racial backgrounds in the U.S. is needed.

## Author Contributions

**Conceptualization:** Kaelia Lynn Saniatan, Smriti Neupane, Gabriela Buccini.

**Data curation:** Chad Cross.

**Formal analysis:** Smriti Neupane, Chad Cross, Gabriela Buccini.

**Funding acquisition:** Gabriela Buccini.

**Methodology:** Kaelia Lynn Saniatan, Smriti Neupane.

**Software:** Chad Cross.

**Supervision:** Chad Cross, Gabriela Buccini.

**Writing – original draft:** Kaelia Lynn Saniatan, Gabriela Buccini.

**Writing – review & editing:** Gabriela Buccini.

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
