## [Decision Letter · Decision Letter 0]

14 Feb 2023

PONE-D-22-35105Socio-demographic, maternal, and infant characteristics associated with pacifier use among six-months old infants in Clark County, Nevada.PLOS ONE

Dear Dr. Buccini,

Thank you for submitting your manuscript to PLOS ONE. After careful consideration, we feel that it has merit but does not fully meet PLOS ONE’s publication criteria as it currently stands. Therefore, we invite you to submit a revised version of the manuscript that addresses the points raised during the review process.

We look forward to receiving your revised manuscript.

Kind regards,

Mona Nabulsi, MD, MS

Academic Editor

PLOS ONE

Reviewers' comments:

Reviewer's Responses to Questions

**Comments to the Author**

1. Is the manuscript technically sound, and do the data support the conclusions?

Reviewer #1: Yes

Reviewer #2: No

2. Has the statistical analysis been performed appropriately and rigorously? 

Reviewer #1: I Don't Know

Reviewer #2: Yes

3. Have the authors made all data underlying the findings in their manuscript fully available?

Reviewer #1: Yes

Reviewer #2: Yes

4. Is the manuscript presented in an intelligible fashion and written in standard English?

Reviewer #1: Yes

Reviewer #2: Yes

5. Review Comments to the Author

Reviewer #1: Very interesting and useful paper, with a good discussion and proposals for helping reducing the use of pacifier in low income families.

Is using any kind of well baby clinic facility linked with a reduction of use of pacifier ?

Is Early use of pacifier linked with social with drawal behavior in infants?

Reviewer #2: This is a well-written paper that addresses an important topic that has not received much attention.

The major issue is the interpretation of the findings related to the multivariable analyses which then has affected the discussion section as well.

Methods

The authors should indicate in the description of the following variable under the measures section that it was within two weeks or after two weeks of life: “The age of introduction was determined by the question, “When did you start giving 137 pacifiers for your baby?” The response options were “No Pacifier Use,” “Within Two Weeks,” or 138 “After Two Weeks.”

Consider revising this sentence in line 155 as it is missing a verb and it is not complete by ending “based on the (28,34):

“…income (≤$50,000/$50-79,999/$80-99,999/≥$100,000). For the household food insecurity screening, the 152 Hunger Vital Sign™, a validated two-question screening tool based on the (28,34).”

It is not clear what the following statement has to do with the description of the second dependent variable in lines 134-135: “The age of pacifier introduction has been associated with the cessation of breastfeeding at six 136 months (33).”

Is there a possible overlap in the observations for the variables exclusive breastfeeding and bottle feeding? Who is the non bottle-feeding group? Does this group include those who are breastfeeding their babies exclusively and those who may be feeding their infant’s solids?

Results:

For table 2: Show the numerical values (results) for the group that did not use a pacifier as well.

Overall, the models should be indicated in the description of the results associated with the multivariable analyses shown in Tables 4 and 5. For example, the independent variable food insecurity in Table 5, was significant only in Model 1, and once other variables entered the subsequent models, this variable was not significant anymore. Overall, the only variable that remained significant after entering all other independent variables was whether the infant was bottle-fed or not. These results need to be emphasized.

The following finding “Pacifier use was independently associated with households with low income (OR (95% CI) 2.06 235 (0.99-4.27))” was not statistically significant. Income level was significantly associated with pacifier use in the first model only, but not in the subsequent models, especially the last model that included all of the variables. Thus, one cannot claim that pacifier use was independently associated with low income households based on that finding.

The results in Table 4 show that those who did not bottle-feed their infants were more likely to use a pacifier and that is the only significant finding in model 5 which contains all of the covariates. Is this finding correct given the results from the bivariate and multinomial analyses? This could be a typo if the authors accidentally switched the reference group for this variable in Table 4.

Again, the authors need to describe the findings from the hierarchical regression modeling in a way that takes into consideration the addition of different sets of variables in each model and indicate the models from which the described results are obtained. Thus, the authors should revise the results described as “Likewise, infants living in food insecure households (RRR (95% CI) 2.53 244 (1.00-6.58)) and mothers who have more than one child (RRR (95% CI) 2.44 (1.11-5.34)) had a higher 245 risk of introducing a pacifier after two weeks.” because they did not remain significant once all variables are entered in the final model.

Discussion:

The reviewer does not believe based on the results that the following conclusion stated in the discussion is fully correct because once all variables are entered into model 5, only the bottle-fed variable remained significant and this means that pacifier use is not independently associated with maternal ethnicity or parity: “Our study identified that pacifier use is independently associated with maternal ethnicity, parity, 251 and bottle feeding among six-month-old living in Clark County, Nevada.”

The same can be said for a number of other findings that did not remain significant in model #5 once all variables were considered in the analyses. These lead to invalid conclusions.

The limitation section should include limitations in regards to the dependent measures such as the usage of pacifiers in the last 24 hours may have excluded a number of mother-infant dyads that used pacifiers for up to a certain age of infancy, etc.

6. PLOS authors have the option to publish the peer review history of their article (what does this mean?). If published, this will include your full peer review and any attached files.

Reviewer #1: **Yes: **Antoine GUEDENEY

Reviewer #2: No

---

## [Author Response · Author response to Decision Letter 0]

16 Mar 2023

To the Editor and reviewers of PLOS ONE:

We thank you for taking the time to review our manuscript and provide detailed and constructive feedback. We reviewed all comments and revised accordingly. We hope they meet your expectations. Please find our point-by-point response below.

and

Response: We reviewed the style requirements and formatted the manuscript accordingly.

b) If there are no restrictions, please upload the minimal anonymized data set necessary to replicate your study findings as either Supporting Information files or to a stable, public repository and provide us with the relevant URLs, DOIs, or accession numbers. For a list of acceptable repositories, please seehttp://journals.plos.org/plosone/s/data-availability#loc-recommended-repositories.

Response: We updated the data availability statement as follow: “A de-identified data set is not possible to provide due to ethical and legal considerations. These sharing restrictions are imposed by the UNLV Institutional Board Review (IRB). The authors declare that a de-identified data set from this study are available upon request directly to Dr. Buccini, assistant professor at UNLV (gabriela.buccini@unlv.edu) and/or to the UNLV IRB (irb@unlv.edu)”

Response: After consulting with the UNLV IRB, we revised the data availability statement and information will be made available upon request.

Review Comments to the Author:

Reviewer #1: Very interesting and useful paper, with a good discussion and proposals for helping reducing the use of pacifier in low income families.

Response: Thank you for this feedback.

1. Is using any kind of well-baby clinic facility linked with a reduction of use of pacifier?

Response: Thank you for this question. The Baby Friendly Hospital Initiative which follows the “Ten Steps to Successful Breastfeeding” have been shown to successfully decrease pacifier use (Venancio et al., 2012). In addition, other evidence-based interventions to increase breastfeeding may lead to reduced pacifier use which includes (i) providing education programs for parents and caregivers through various channels such as prenatal classes, postnatal support groups, and online resources (Rocha et al., 2011), (ii) increasing access to community-based initiatives that offer perinatal support through peer counselor (Chapman et al., 2010) and lactation consultants (Haroon et al.,2013), (iii) training of health providers can help ensure that parents receive consistent messages on pros and cons of pacifier use, and (iv) promoting regulatory policies such as restrictions of predatory marketing of pacifiers (Pérez-Escamilla et al., 2023; Rollins et al., 2023). Many of these recommendations are embedded into the Baby Friendly Hospital Initiative which follows the “Ten Steps to Successful Breastfeeding”. While our study did not find an association between place of birth and pacifier use, potentially due to the low coverage of births occurred in Baby Friendly Hospitals in Clark County (19.9% vs 27.0% in the U.S (Wright, 2023)); it is important to note that scaling up Baby Friendly Hospital Initiative has been associated with increased rates of exclusive breastfeeding and decreased rates of pacifier use (Venancio et al., 2012). We added this information into the discussion section lines 427-44

References:

1. Chapman, D. J., Morel, K., Anderson, A. K., Damio, G., & Pérez-Escamilla, R. (2010). Breastfeeding peer counseling: from efficacy through scale-up. Journal of human lactation : official journal of International Lactation Consultant Association, 26(3), 314–326. https://doi.org/10.1177/0890334410369481

2. Haroon, S., Das, J. K., Salam, R. A., Imdad, A., & Bhutta, Z. A. (2013). Breastfeeding promotion interventions and breastfeeding practices: a systematic review. BMC public health, 13 Suppl 3(Suppl 3), S20. https://doi.org/10.1186/1471-2458-13-S3-S20

3. Pérez-Escamilla, R., Tomori, C., Hernández-Cordero, S., Baker, P., Barros, A. J. D., Bégin, F., Chapman, D. J., Grummer-Strawn, L. M., McCoy, D., Menon, P., Neves, P. a. R., Piwoz, E., Rollins, N., Victora, C. G., & Richter, L. (2023). Breastfeeding: crucially important, but increasingly challenged in a market-driven world. The Lancet, 401(10375), 472–485. https://doi.org/10.1016/s0140-6736(22)01932-8

4. Rocha, C. C., Verga, K. E., Sipsma, H., Larson, I. A., Phillipi, C. A., & Kair, L. R. (2020). Pacifier Use and Breastfeeding: A Qualitative Study of Postpartum Mothers. Breastfeeding Medicine, 15(1), 24-28.

https://doi.org/10.1089/bfm.2019.0174

5. Rollins, N., Piwoz, E., Baker, P., Kingston, G., Mabaso, K. M., McCoy, D., Neves, P. a. R., Pérez-Escamilla, R., Richter, L., Russ, K., Sen, G., Tomori, C., Victora, C. G., Zambrano, P., & Hastings, G. (2023). Marketing of commercial milk formula: a system to capture parents, communities, science, and policy. The Lancet, 401(10375), 486–502. https://doi.org/10.1016/s0140-6736(22)01931-6

6. Venancio, S. I., Saldiva, S. R., Escuder, M. M., & Giugliani, E. R. (2012). The Baby-Friendly Hospital Initiative shows positive effects on breastfeeding indicators in Brazil. Journal of epidemiology and community health, 66(10), 914–918. https://doi.org/10.1136/jech-2011-200332

7. Wright, R. (2023, February 3). Baby-Friendly USA - About. Baby-Friendly USA. https://www.babyfriendlyusa.org/about/#:~:text=In%202007%2C%20less%20than%203,and%20they%20continue%20to%20rise.

2. Is Early use of pacifier linked with social withdrawal behavior in infants?

Response: Thanks for this question. While there is no evidence on pacifier use and infant social withdrawal, pacifier use has been associated to speech problems (Shotts et al., 2008), lower social interactions (Baraca., 2021), and lower levels of intelligence. A recent analysis of a birth cohort study found a strong association between pacifier use and lower intelligence quotient at 6 years (Giugliani et al., 2021). One possible hypothesis is that children using a pacifier, especially those who use it more intensely, are less stimulated affecting social interaction skills and may lose a critical window for optimal early childhood development exacerbating disparities early in life. 

We added this information into the discussion section lines 413-418

References:

1. Barca, L. (2021). Toward a speech-motor account of the effect of Age of Pacifier Withdrawal. Journal of Communication Disorders, 90, 106085. https://doi.org/10.1016/j.jcomdis.2021.106085

2. Giugliani, E. R. J., Gomes, E., Santos, I. S., Matijasevich, A., Camargo-Figuera, F. A., & Barros, A. J. D. (2021). All day-long pacifier use and intelligence quotient in childhood: A birth cohort study. Paediatric and perinatal epidemiology, 35(4), 511–518. https://doi.org/10.1111/ppe.12752 \\

3. Shotts LL, McDaniel M, Neeley RA. The Impact of Prolonged Pacifier Use on Speech Articulation: A Preliminary Investigation. Contemporary Issues in Communication Science and Disorders. 2008 Mar;35(Spring):72–5.

Reviewer #2: This is a well-written paper that addresses an important topic that has not received much attention

Response: Thank you for this feedback.

3. The major issue is the interpretation of the findings related to the multivariable analyses which then has affected the discussion section as well.

Response: We revised our data analysis section to clarify that we used a hierarchical modelling data analysis approach (Victora et al., 1997) which is particularly appropriate for use in studying determinants of childhood health outcomes such as breastfeeding (Nascimento et al., 2012) and pacifier use (Buccini et al., 2014), as in the case of our analysis. Following the hierarchical modeling theory (Victora et al., 1997) we revised the paragraph in lines 191-199 to detail the modeling process as well as interpretation of findings. Further elaboration on the findings interpretations were made to both the results (lines 302-204 and 318-322) and discussion (lines 476-484) sections of the manuscript. 

References:

1. Nascimento, E. N., Leone, C., Abreu, L. C. de, & Buccini, G. (2021). Determinants of exclusive breast-feeding discontinuation in southeastern Brazil, 2008–2013: A pooled data analysis. Public Health Nutrition, 24(10), 3116–3123. https://doi.org/10.1017/S1368980020003110

2. Victora, C. G., Huttly, S. R., Fuchs, S. C., & Olinto, M. T. (1997). The role of conceptual frameworks in epidemiological analysis: A hierarchical approach. International Journal of Epidemiology, 26(1), 224–227. https://doi.org/10.1093/ije/26.1.224

3. Buccini, G. dos S., Benício, M. H. D., & Venancio, S. I. (2014). Determinants of using pacifier and bottle feeding. Revista de Saúde Pública, 48(4), 571–582. https://doi.org/10.1590/S0034-8910.2014048005128

Methods:

4. The authors should indicate in the description of the following variable under the measures section that it was within two weeks or after two weeks of life: “The age of introduction was determined by the question, “When did you start giving 137 pacifiers for your baby?” The response options were “No Pacifier Use,” “Within Two Weeks,” or 138 “After Two Weeks.”

Response: Clarification of the age of the infant within the response options were added in lines 136-137 and 165. 

5. Consider revising this sentence in line 155 as it is missing a verb and it is not complete by ending “based on the (28,34):

“…income (≤$50,000/$50-79,999/$80-99,999/≥$100,000). For the household food insecurity screening, the 152 Hunger Vital Sign™, a validated two-question screening tool based on the (28,34).”

Response: Done. Lines 154-155.

6. It is not clear what the following statement has to do with the description of the second dependent variable in lines 134-135: “The age of pacifier introduction has been associated with the cessation of breastfeeding at six 136 months (33).”

Response: We removed this sentence from the paragraph.

7. Is there a possible overlap in the observations for the variables exclusive breastfeeding and bottle feeding? Who is the non bottle-feeding group? Does this group include those who are breastfeeding their babies exclusively and those who may be feeding their infant’s solids?

Response: Bottle-feeding group was defined as the children who are fed with any liquid (including breast milk) or semi-solid food from a bottle with nipple/teat, as recommended by the World Health Organization. The definition of bottle feeding was included in the methods section lines 177-179.

Results:

8. For table 2: Show the numerical values (results) for the group that did not use a pacifier as well.

Response: Done. Lines 294-295.

9. Overall, the models should be indicated in the description of the results associated with the multivariable analyses shown in Tables 4 and 5. For example, the independent variable food insecurity in Table 5, was significant only in Model 1, and once other variables entered the subsequent models, this variable was not significant anymore. Overall, the only variable that remained significant after entering all other independent variables was whether the infant was bottle-fed or not. These results need to be emphasized.

Response: We updated the results section to reflect the hierarchical modeling approach by indicating in which level and model variables were significant. These changes are reflected in lines 302-304, 318-322 and Tables 4 and 5. As indicated above, for further clarification we edited both methods (lines 199-214) and discussion (lines 480-488) sections of the manuscript.

10. The following finding “Pacifier use was independently associated with households with low income (OR (95% CI) 2.06 235 (0.99-4.27))” was not statistically significant. Income level was significantly associated with pacifier use in the first model only, but not in the subsequent models, especially the last model that included all of the variables. Thus, one cannot claim that pacifier use was independently associated with low income households based on that finding.

Response: Thank you for noticing this error. The association between low household income and pacifier use was removed from this statement as it did not withhold statistical significance.

11. The results in Table 4 show that those who did not bottle-feed their infants were more likely to use a pacifier and that is the only significant finding in model 5 which contains all of the covariates. Is this finding correct given the results from the bivariate and multinomial analyses? This could be a typo if the authors accidentally switched the reference group for this variable in Table 4.

Response: Thank you for your careful review. The reference group for bottle feeding in Table 4 was accidently switched. The true reference is now in the correct place. 

12. Again, the authors need to describe the findings from the hierarchical regression modeling in a way that takes into consideration the addition of different sets of variables in each model and indicate the models from which the described results are obtained. Thus, the authors should revise the results described as “Likewise, infants living in food insecure households (RRR (95% CI) 2.53 244 (1.00-6.58)) and mothers who have more than one child (RRR (95% CI) 2.44 (1.11-5.34)) had a higher 245 risk of introducing a pacifier after two weeks.” because they did not remain significant once all variables are entered in the final model.

Response: As described above the hierarchical modeling process as well as interpretation of findings were clarified in the data analysis section (lines 199-214). Based on the hierarchical modeling approach, food insecure households and mothers with more than one child were statistically significant at level 2 model. This has been added to the result section lines 318-322 . 

Discussion:

13. The reviewer does not believe based on the results that the following conclusion stated in the discussion is fully correct because once all variables are entered into model 5, only the bottle-fed variable remained significant and this means that pacifier use is not independently associated with maternal ethnicity or parity: “Our study identified that pacifier use is independently associated with maternal ethnicity, parity, 251 and bottle feeding among six-month-old living in Clark County, Nevada.” The same can be said for a number of other findings that did not remain significant in model #5 once all variables were considered in the analyses. These lead to invalid conclusions.

Response: Thank you for this recommendation. Following the hierarchical modeling theory described in the methods section, we opted to edit the paragraph including the level model in which the variable was significantly associated (340-343). In addition, we added a statement to discuss the limitations of the hierarchical model in the discussion (lines 443-451)

14. The limitation section should include limitations in regards to the dependent measures such as the usage of pacifiers in the last 24 hours may have excluded a number of mother-infant dyads that used pacifiers for up to a certain age of infancy, etc.

Response: Thank you for this suggestion. This was elaborated on in the discussion (lines 452-479)

---

## [Decision Letter · Decision Letter 1]

17 Apr 2023

Socio-demographic, maternal, and infant characteristics associated with pacifier use among six-months old infants in Clark County, Nevada.

PONE-D-22-35105R1

Dear Dr. Buccini,

We’re pleased to inform you that your manuscript has been judged scientifically suitable for publication and will be formally accepted for publication once it meets all outstanding technical requirements.

Kind regards,

Mona Nabulsi, MD, MS

Academic Editor

PLOS ONE

Reviewers' comments:

Reviewer's Responses to Questions

**Comments to the Author**

1. If the authors have adequately addressed your comments raised in a previous round of review and you feel that this manuscript is now acceptable for publication, you may indicate that here to bypass the “Comments to the Author” section, enter your conflict of interest statement in the “Confidential to Editor” section, and submit your "Accept" recommendation.

Reviewer #2: All comments have been addressed

2. Is the manuscript technically sound, and do the data support the conclusions?

Reviewer #2: Yes

3. Has the statistical analysis been performed appropriately and rigorously? 

Reviewer #2: Yes

4. Have the authors made all data underlying the findings in their manuscript fully available?

Reviewer #2: Yes

5. Is the manuscript presented in an intelligible fashion and written in standard English?

Reviewer #2: Yes

6. Review Comments to the Author

Reviewer #2: Overall, the authors did a good job in addressing the comments.

The paper has improved considerably.

7. PLOS authors have the option to publish the peer review history of their article (what does this mean?). If published, this will include your full peer review and any attached files.

Reviewer #2: No

---

## [Editor Report · Acceptance letter]

19 Apr 2023

PONE-D-22-35105R1 

Socio-demographic, maternal, and infant characteristics associated with pacifier use among six-months old infants in Clark County, Nevada. 

Dear Dr. Buccini:

I'm pleased to inform you that your manuscript has been deemed suitable for publication in PLOS ONE. Congratulations! Your manuscript is now with our production department. 

Kind regards, 

on behalf of

Dr. Mona Nabulsi 

Academic Editor

PLOS ONE